# CAST: Time-Varying Treatment Effects with Application to Chemotherapy and Radiotherapy on Head and Neck Squamous Cell Carcinoma

## Abstract

Causal machine learning (CML) enables individualized estimation of treatment effects, offering critical advantages over traditional correlation-based methods. However, existing approaches for medical survival data with censoring such as causal survival forests estimate effects at fixed time points, limiting their ability to capture dynamic changes over time. We introduce Causal Analysis for Survival Trajectories (CAST), a novel framework that models treatment effects as continuous functions of time following treatment. By combining parametric and non-parametric methods, CAST overcomes the limitations of discrete time-point analysis to estimate continuous effect trajectories. Using the RADCURE dataset [1] of 2,651 patients with head and neck squamous cell carcinoma (HNSCC) as a clinically relevant example, CAST models how chemotherapy and radiotherapy effects evolve over time at the population and individual levels. By capturing the temporal dynamics of treatment response, CAST reveals how treatment effects rise, peak, and decline over the follow-up period, helping clinicians determine when and for whom treatment benefits are maximized. This framework advances the application of CML to personalized care in HNSCC and other life-threatening medical conditions.

## 1 Introduction

**Methodological gap:** A critical limitation in most traditional statistical and machine learning (ML) methods applied to clinical outcomes data is their correlational nature. These methods are designed to identify associations between variables but are not equipped to answer causal questions, which are central to understanding treatment effects. In clinical research, the key questions—such as how a treatment impacts survival or which patients benefit most—are inherently causal. However, correlational approaches cannot disentangle confounding factors or provide interpretable estimates of causal relationships, leaving a significant methodological gap [2, 3].

Causal machine learning (CML) offers a promising solution by explicitly modeling causal effects rather than associations. CML is rapidly advancing, providing tools to estimate individualized and subgroup-specific treatment effects [4]. However, current causal forest methods adapted for survival data fall short in one crucial aspect: they estimate treatment effects only at discrete time horizons after treatment [5, 6, 7]. This approach fails to capture the continuous evolution of treatment effects over time, limiting their ability to address dynamic clinical questions.

**Proposed approach:** Our novel method, CAST (Causal Analysis for Survival Trajectories), fills this gap by extending causal survival forests to provide continuous treatment effect estimates as a function of time after treatment. CAST combines parametric and non-parametric techniques to

model the temporal dynamics of treatment effects, offering a more nuanced and clinically relevant understanding of how treatments impact outcomes over time [8, 9]. We build upon previous work by Shuryak et al. [10] to extend it to chemotherapy and continuous-time causal modeling. By addressing this methodological gap, CAST enables clinicians to answer the causal questions that matter most for personalized care and evidence-based decision-making. While traditional approaches estimate treatment effects at discrete time points [11, 12], CAST provides a continuous mathematical framework, analyzing how treatment benefits evolve over the entire follow-up period. In the context of cancer therapy, this is key: biological responses unfold through complex temporal dynamics that include initial tumor control followed by potential diminishing returns due to repopulation, late toxicities, and other factors [13, 14, 15].

**Clinical motivation:** We evaluate CAST in the context of head and neck squamous cell carcinoma (HNSCC), where treatment responses evolve over time and vary across patient subgroups. HNSCC, ranked as the seventh most prevalent cancer worldwide, includes malignancies of the oral cavity, pharynx, larynx, and other surrounding regions of the head and neck. With incidence rates rising rapidly, HNSCC is projected to increase nearly 30 % annually by 2030 [16]. Historically, most HNSCC cases were attributed to excessive alcohol and tobacco use, with heavy exposure increasing risk by up to 40-fold [17]. However, the past two decades have seen an increase in human papillomavirus-related (HPV) cases, and HPV-associated HNSCC is expected to surpass tobacco and alcohol induced tumors in the next five years. This has caused a shift in the demographic profile of HNSCC patients: HPV-related cases tend to occur among younger populations (<65), particularly in men [18, 19, 20].

To treat HNSCC, clinicians use combinations of surgery, chemotherapy, and radiation. Intensity-modulated radiation therapy (IMRT) has become the standard of care for its precision in targeting tumors while sparing healthy tissue [21, 22]. Studies show that IMRT's impact on patient quality of life follows a time-varying trajectory, with distinct peaks of symptom burden and phases of recovery [23, 24]. In a similar vein, chemotherapy—involving agents that disrupt the DNA of rapidly dividing cells—follows a variable treatment timeline [25]. Patients are often advised that responses can differ widely based on individual factors, with effects emerging gradually and no fixed timeline for when benefits or side effects will manifest [26, 27].

The challenge of analyzing radiation therapy and chemotherapy outcomes lies not just in the complexity of the treatment itself, but in the multitude of factors that influence both treatment assignment and patient response. Traditional correlation-based analyses can mask important causal relationships, leading to suboptimal treatment decisions. To the best of our knowledge, CAST represents the first causal machine learning framework to explicitly model how chemotherapy benefits for patient survival rise and fall over the entire follow-up period.

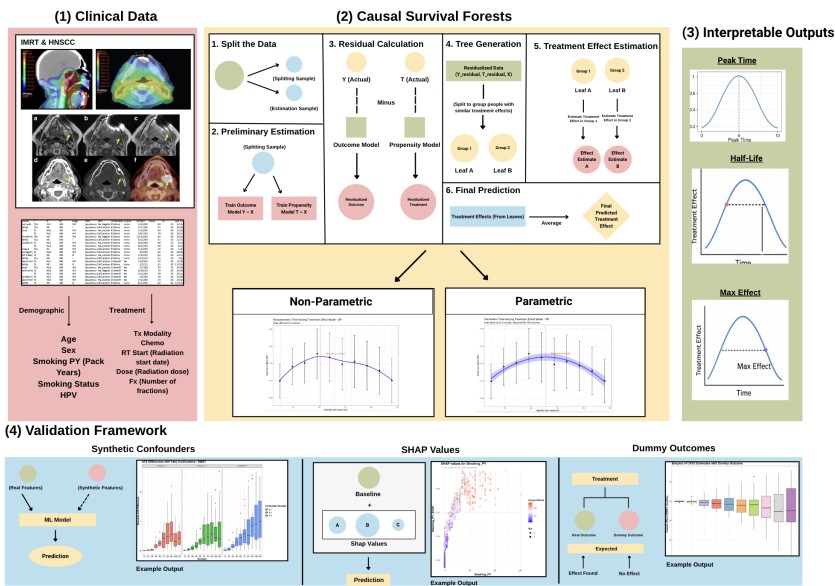

Figure 1: Overview of the CAST framework

**Modeling philosophy:** Our novel machine learning approach leverages causal survival forests to handle high-dimensional data while automatically discovering treatment effect heterogeneity, and differs from conventional survival analysis methods by focusing explicitly on estimating causal treatment effects while accounting for confounding factors through careful propensity score modeling. As demonstrated in Figure 1, the framework incorporates both parametric modeling to capture characteristic rise and fall patterns of chemotherapy effects and non-parametric approaches to reveal subtle inflection points corresponding to biological phase transitions in treatment response.

We implemented a variety of methods to ensure a robust assessment of the data and to verify key causal inference assumptions, such as overlap (positivity) and no unmeasured confounding (ignorability). We used elastic net logistic regression with repeated k-fold cross-validation to estimate propensity scores for chemotherapy. Patients with scores outside the range [0.1, 0.9] were trimmed to ensure overlap between treatment groups. We then conducted refutation tests, including dummy outcome and negative control analyses, to assess the robustness of our causal effect estimates. Using SHapley Additive exPlanations (SHAP) values, we generated interpretable insights into how patient and disease characteristics impact treatment outcomes, allowing for practical application in clinical settings.

**Significance:** This research applies causal survival forests to identify how patient and disease characteristics—like age and HPV status—influence treatment effectiveness. By combining advanced causal inference with CAST's temporal modeling, we can determine not just who benefits most from chemotherapy, but also when these benefits peak and fade. This temporal insight is key for designing targeted interventions and optimizing outcomes in HNSCC. CAST also demonstrates the broader potential of integrating machine learning into personalized cancer care.

**Our contributions are as follows:**

- **CAST is, to our knowledge, the first framework** to unify causal survival forests with *parametric* and *non-parametric* models for estimating **continuous-time treatment effects**, offering a new paradigm for temporal causal inference in survival analysis.
- CAST produces clinically interpretable metrics such as *peak effect time*, *maximum benefit*, and *effect half-life*, enabling richer understanding of treatment response dynamics.
- We introduce a rigorous validation framework incorporating *propensity score modeling*, *dummy outcome tests*, *synthetic tests*, and *SHAP-based heterogeneity analysis*.
- We apply CAST to a large real-world chemotherapy and radiotherapy dataset (RADCURE), uncovering actionable insights into when and for whom treatment benefits peak and decline.

## 2   Related Work

**Clinical predictors of treatment response:** Numerous studies have shown that treatment response in HNSCC patients is highly heterogeneous, influenced by clinical and demographic factors such as HPV status, gender, and disease stage. For instance, HPV-positive HNSCC—more common among younger patients—tends to be more sensitive to treatment and is associated with more favorable survival outcomes compared to HPV-negative disease [28]. Historically, studies that group patients by their clinical characteristics reveal significant variation in survival [29, 30]. These findings motivate the need for methods that can model treatment effect heterogeneity as well as average treatment effects (ATE)—an aim CAST directly addresses.

**Predictive survival models:** Traditional models such as the Cox proportional hazards model assume proportional hazards and constant treatment effects over time, limiting their flexibility in capturing nonlinear and time-varying dynamics [31, 32]. More flexible models—including random survival forests (RSF), deep survival models (e.g., DeepSurv), and Bayesian additive regression trees (BART)—have demonstrated improved risk prediction performance [33], notably in applications such as cervical cancer survival [34, 35]. However, these models are fundamentally *predictive*, not causal—they estimate outcome risk without isolating treatment effects or correcting for confounding unless explicitly adapted with causal modeling components.

**Causal inference for survival analysis:** Recent advances in causal machine learning have introduced methods designed to estimate individualized treatment effects (ITEs) from observational time-to-event data [36, 37]. These include meta-learners (e.g., T-learner, S-learner) [38], G-formula-based two-learners [39], double robust estimators (e.g., AIPCW and AIPTW) [40], and causal survival forests

[41]. While these causal approaches are advantageous for treatment effect estimation compared with traditional survival analysis, they typically estimate effects at discrete time points, limiting their ability to model how treatment responses evolve continuously throughout follow-up [42, 43].

**Modeling time-varying treatment effects:** In oncology, treatment effects often unfold through distinct biological phases—initial tumor control, plateauing benefit, and eventual decline due to late toxicities or tumor repopulation [44, 45]. Studies have revealed that the prognostic influence of covariates such as age, race, and sex changes over the follow-up period [46, 47]. However, most existing methods either assume constant effects or treat follow-up intervals independently. CAST addresses this gap by modeling treatment effects as continuous functions of time. By integrating both parametric (e.g., quadratic fits) and non-parametric (e.g., smoothing splines) components, CAST captures biologically grounded patterns in treatment efficacy over time. Unlike previous approaches, CAST provides a unified, continuous-time framework that reveals the full temporal trajectory of treatment response, enabling more precise and interpretable causal insights.

# 3 Methodology

**Problem Formulation:** We address the challenge of estimating time-varying treatment effects in survival analysis, specifically focusing on how the impact of medical interventions evolves over time. Let $\mathcal{D} = \{(X_i, W_i, T_i, \delta_i)\}_{i=1}^n$ represent our dataset where:

- $X_i \in \mathbb{R}^p$ is a vector of covariates for subject $i$
- $W_i \in \{0, 1\}$ is the treatment indicator
- $T_i$ is the observed survival time (either event time or censored time)
- $\delta_i$ is the event indicator (1 if event observed, 0 if censored)

The causal survival forest method is a powerful tool for estimating average and subgroup-specific treatment effects for survival outcomes, but it estimates the effects only at specific discrete times after treatment. This fails to capture the continuous temporal evolution of treatment responses, particularly in contexts like radiation therapy and chemotherapy where biological effects can substantially rise and fall over time.

## 3.1 Causal Machine Learning Framework

Our approach uses a CML framework to isolate treatment effects beyond traditional correlational methods. While conventional machine learning identifies correlations between variables, CML allows us to understand the causal impact of interventions [48]. This distinction is fundamental to our study: our goal is not just to predict outcomes but to dissect how treatments shape survival outcomes across patient subgroups.

Given the observational non-randomized nature of our clinical data, we rely on the following assumptions:

- **Unconfoundedness**: Treatment assignment is independent of potential outcomes conditional on observed covariates (also called ignorability or no unmeasured confounding)
- **Positivity (Overlap)**: Every subject has a non-zero probability of receiving each treatment
- **Consistency**: A subject's observed outcome under their received treatment equals their potential outcome for that treatment
- **Non-interference**: One subject's treatment does not affect another subject's outcome

To address selection bias in observational data, we performed propensity score modeling using elastic net logistic regression: $\hat{e}(X) = P(W = 1|X)$ with hyperparameters optimized through 10-fold cross-validation. Patients with extreme propensity scores (outside $[0.10, 0.90]$) are trimmed to ensure overlap between treatment groups. See Appendix C.1 for balance diagnostics.

## 3.2 CAST: Causal Analysis for Survival Trajectories

The theoretical foundation of CAST rests on modeling the effect trajectory as a function of time. Our target estimand is the conditional average treatment effect (CATE) at time $t$, given covariates $X$:

$$\tau(x,t) = \mathbb{E}[Y(1,t) - Y(0,t) \mid X = x] \tag{1}$$

where $Y(w,t)$ represents the potential outcome at time $t$ under treatment $w$, and $x$ denotes an individual's covariates. We consider two types of time-varying estimands: the difference in restricted mean survival time (RMST) and the difference in survival probability (SP) between treatment groups. Unlike prior methods that estimate effects at fixed time points, CAST models treatment effects as smooth functions of time. We use a smoothing spline to estimate the continuous effect trajectory, and a quadratic fit to derive interpretable metrics.

### 3.2.1 Parametric Modeling Component

Our parametric modeling component employs a quadratic function: $\tau(t) = \beta_0 + \beta_1 t + \beta_2 t^2$ to capture the rise and fall of treatment effects. The parameters are estimated using weighted least squares:

$$\min_{\beta_0, \beta_1, \beta_2} \sum_t w(t)(\hat{\tau}(t) - (\beta_0 + \beta_1 t + \beta_2 t^2))^2 \tag{2}$$

where $w(t) = 1/\sigma^2(t)$ are weights based on the variance of the effect estimates at each timepoint. This approach yields clinically interpretable parameters, including the peak effect time ($t_{\text{peak}} = -\beta_1/2\beta_2$), the maximum effect magnitude ($\tau(t_{\text{peak}})$), and the treatment effect half-life, defined as the time it takes for the effect to diminish by 50% from its peak.

These parameters directly quantify key clinical aspects of the treatment response: when the maximum benefit occurs, how large that benefit is, and how quickly it diminishes—information critical for clinical decision-making that traditional methods cannot provide. See Appendix C.3 for fitted coefficients and summary statistics from the parametric model.

---

**Algorithm 1** CAST-PARAMETRIC

---

1: **Input:** Horizons $\mathcal{H}$, ATEs $\{\hat{\tau}_h\}$, SEs $\{\hat{\sigma}_h\}$
2: **Output:** Temporal function $\hat{\tau}(t)$, peak time $t^*$, half-life $\lambda$
3: $\mathcal{W} \leftarrow \{w_h = 1/\hat{\sigma}_h^2\}$ ▷ Inverse-variance weights
4: $\hat{\tau}(t) \leftarrow$ FITQUADRATICMODEL$(\mathcal{H}, \hat{\tau}, \mathcal{W})$
5: $\beta_1, \beta_2 \leftarrow$ coefficients from fit
6: **if** $\beta_2 \neq 0$ **then**
7: $\quad t^* \leftarrow -\beta_1/(2\beta_2)$ ▷ Time of peak effect
8: $\quad \lambda \leftarrow$ SOLVE$(\hat{\tau}(t^* + \lambda) = \hat{\tau}(t^*)/2)$
9: **else**
10: $\quad t^*, \lambda \leftarrow$ NA ▷ Degenerate case
11: **end if**
12: **return** $\hat{\tau}(t), t^*, \lambda$

---

**CAST-Parametric:** This algorithm models treatment effects over time using a weighted quadratic fit to the estimated ATEs across discrete horizons. Inverse-variance weighting emphasizes more confident estimates. The peak effect time is derived analytically, while the half-life is computed by numerically solving for the point where the curve falls to half its maximum. This approach yields interpretable summaries of treatment dynamics, aligning with radiobiological phenomena such as delayed benefit and diminishing returns.

### 3.2.2 Non-parametric Modeling Component

Our non-parametric component employs cross-validated smoothing splines:

$$\tau(t) = g(t), \quad \text{where} \quad g = \arg\min_f \left\{ \sum_t w(t) \left(\hat{\tau}(t) - f(t)\right)^2 + \lambda \int f''(t)^2 \, dt \right\} \tag{3}$$

where $\lambda$ is selected via cross-validation. This approach adapts to the data without imposing a predetermined functional form, revealing subtle inflection points in the effect trajectory that correspond to biological phase transitions in the treatment response.

We calculate the first and second derivatives of the fitted spline to identify key features of the treatment effect trajectory: local maxima and minima where $g'(t) = 0$, acceleration and deceleration phases based on sign changes in $g''(t)$, and inflection points where $g''(t) = 0$.

The non-parametric model complements the parametric fit by capturing complex, less predictable patterns—especially during later follow-up periods, when biological processes like accelerated repopulation and late toxicities may cause deviations from the smooth quadratic trend.

**Algorithm 2** CAST-NONPARAMETRIC

1: **Input:** Horizons $\mathcal{H}$, ATEs $\{\hat{\tau}_h\}$, SEs $\{\hat{\sigma}_h\}$
2: **Output:** Spline $\hat{\tau}(t)$, peak $t^*$, inflections $\{t_i\}$
3: $\mathcal{W} \leftarrow \{w_h = 1/\hat{\sigma}_h^2\}$
4: $\hat{\tau}(t) \leftarrow \text{FITSPLINE}(\mathcal{H}, \hat{\tau}, \mathcal{W})$
5: $D_1(t), D_2(t) \leftarrow$ first and second derivatives of $\hat{\tau}(t)$
6: $t^* \leftarrow \text{ARGMAX}(\hat{\tau}(t))$        ▷ Peak effect
7: $\{t_i\} \leftarrow \text{ZEROCROSSINGS}(D_2(t))$      ▷ Inflection points
8: **if** $t^*$ not in $[\min(\mathcal{H}), \max(\mathcal{H})]$ **then**
9:     $t^* \leftarrow \text{NA}$
10: **end if**
11: **return** $\hat{\tau}(t), t^*, \{t_i\}$

**CAST-Nonparametric:** This algorithm fits a smoothing spline to the estimated treatment effects across time using inverse-variance weights. It computes the first and second derivatives of the spline to identify key dynamics: the peak effect time via the curve's global maximum and biological phase transitions via inflection points. This method captures delayed and non-monotonic effect trajectories often missed by parametric models, reflecting immune response, tissue adaptation, or timing heterogeneity.

CAST-Parametric and CAST-Nonparametric offer complementary modeling capabilities. The parametric method provides interpretable summary statistics such as peak effect timing and half-life, which are clinically intuitive and useful for hypothesis testing under smooth treatment dynamics. In contrast, the spline-based approach relaxes these assumptions and flexibly captures nonlinear, delayed, or multi-phase effects. Together, these models allow us to evaluate the robustness of temporal patterns and support a wide range of clinical interpretations.

**Theoretical Guarantees:** See Appendix A for theorem statements establishing consistency of CAST estimators and identifiability of time-varying treatment effects under standard causal assumptions.

# 4 Experiments

**Dataset:** We use the RADCURE observational dataset from The Cancer Imaging Archive (TCIA), a publicly accessible resource on multiple types of cancer. The dataset spans from 2005 to 2017 and contains clinical, demographic, and treatment metadata for 3,346 patients. We select 2,651 patients with pathologically confirmed HNSCC and a defined tumor site. While the dataset primarily focuses on oropharyngeal cancer, it also includes laryngeal, nasopharyngeal, and hypopharyngeal cases. The binary treatment variable used in CAST is chemotherapy (yes/no) with radiotherapy covariates.

**Preprocessing:** We filtered incomplete profiles and standardized continuous variables for comparability. We used radiotherapy data—dose/fraction, number of fractions, and total radiation treatment time duration in days—to calculate Biologically Effective Dose (BED) values, applying both dose-independent (DI) and dose-dependent (DD) models with established radiobiological parameters [10]. We then partitioned the dataset into training (75%) and testing (25%) sets, maintaining consistent event rates across both subsets for unbiased evaluation of treatment effects. See Appendix B for more on data preprocessing and computing resources.

**Propensity Score Modeling:** To address selection bias, we used elastic net logistic regression to estimate the likelihood of a person receiving treatment, based on their characteristics. Hyperparameters were optimized through 10-fold cross-validation: elastic net mixing parameter $\alpha \in [0.01, 0.99]$ and regularization parameter $\lambda$ chosen from a grid of 100 values. Propensity score distributions were assessed through both Pearson and Spearman correlation matrices ($\alpha = 0.05$, Bonferroni-corrected) and visualized using kernel density estimation. Patients with scores outside $[0.10, 0.90]$ were trimmed to ensure overlap, with sensitivity analyses conducted at thresholds $\{0.01, 0.03, 0.05, 0.07, 0.10\}$.

### Implementation & Heterogeneity Analysis

We used causal survival forests with Nelson-Aalen estimation to handle right-censoring, estimating treatment effects over $12, 24, \ldots, 120$ months post-treatment. Our forest was constructed with 5,000

trees to ensure robust estimation of heterogeneous effects across the patient population. Sensitivity analyses using different numbers of trees showed similar results.

For each time horizon, we independently trained a causal forest model using the training dataset, with covariates properly standardized and propensity scores incorporated through doubly-robust estimation. The forests were configured with tuning parameters selected through cross-validation, including minimum node size, split regularization, and sampling fraction. Prediction uncertainty was quantified through the infinitesimal jackknife method, providing variance estimates for each individual treatment effect. This approach allowed us to capture both average treatment effects and their heterogeneity across different patient subgroups at each follow-up time point, while properly accounting for the right-censoring inherent in survival data [49, 50].

Treatment effect heterogeneity was analyzed using approximate SHAP values calculated via Monte Carlo sampling with 1,000 iterations and a convergence threshold of $\epsilon = 0.01$. The SHAP values were normalized such that $\sum_i \text{SHAP}_i$ corresponds to the difference between the individual and mean model predictions. This approach revealed which patient characteristics most strongly influenced treatment response, with HPV status and smoking history emerging as particularly important predictors. We visualized the relationship between feature values and their SHAP contributions to identify subgroups with differential treatment benefits.

**Validation Methods**

We implemented several validation strategies as refutation tests for the causal effect estimates in our experiments. For each test, we computed summary statistics (mean, standard deviation, max deviation) to assess model robustness, using a consistent 5,000-tree specification and random seeds for reproducibility.

**Dummy Outcome Tests:** We shuffled treatment assignments and outcome times across 20 repetitions for each time horizon (12-120 months), generating a null distribution to assess false positive rates. Boxplots confirmed the null hypothesis centered around zero, showing that the causal effect estimates for each horizon were centered around zero as expected. The variance of these estimates increased with increasing horizon time due to the decreasing number of patients remaining at risk at longer times. The results suggested good reliability of the estimates for times $\leq 60$ months.

**Sensitivity to Additional Covariates:** We introduced synthetic covariates with varying signal strengths of correlation with treatment assignment (0.1, 0.3, 0.5) that were unrelated to both treatment assignment and outcome, in order to assess the sensitivity of treatment effect estimates to irrelevant/spurious variables.

**Negative Control Tests:** Irrelevant binary treatments were randomly assigned to ensure the model did not detect spurious effects. Treatment effects for these were zero across all time horizons.

**Robustness to Irrelevant Features:** Five random noise variables were added, and changes in treatment effect estimates and feature importance were monitored to ensure no significant impact.

# 5 Results

We present empirical results of CAST on the RADCURE dataset, focusing on time-varying treatment effects, patient-level heterogeneity, and robustness validation. As shown in Figure 2 below, CAST reveals a non-monotonic trajectory in chemotherapy benefit: survival gains increase early post-treatment, plateau in the mid-term, and gradually decline thereafter. Both the parametric and non-parametric models suggest a peak in benefit between 50 and 65 months, though the effect trajectory remains relatively stable during this period. These trends indicate that chemotherapy is most impactful in the first few years post-treatment, with gradual tapering over time. This is potentially due to recurrence, long-term toxicity, or competing risks. On the testing set, chemotherapy increased survival probability by $15.2 \pm 6.0\%$ at 3 years and $15.0 \pm 6.7\%$ at 5 years, with RMST gains of $3.6 \pm 1.4$ and $7.1 \pm 2.6$ months, respectively.

**Individualized effect distributions:** Treatment effect estimates showed notable variation across patients. While most individuals experienced positive effects, CAST identified a long right tail of high responders and a small subset with near-zero or negative effects. However, some of this variation may

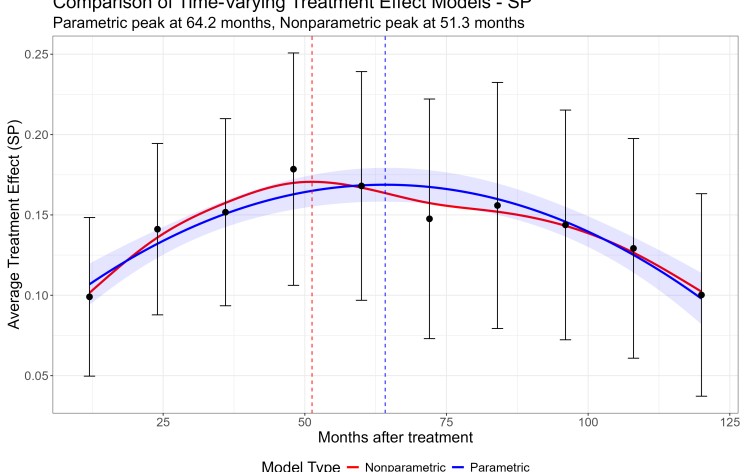

Figure 2: Comparison of time-varying treatment effect models using CAST. The red curve shows the parametric estimate with 95% CIs; the blue curve shows the non-parametric spline. Black dots denote average treatment effects ± standard errors on the survival probability scale.

reflect unmeasured confounding or estimation noise rather than true heterogeneity. These patterns highlight the potential for personalized models in survival-based decision-making.

**Subgroup variation:** To identify drivers of treatment heterogeneity, we computed Pearson and Spearman correlation matrices between clinical covariates, SHAP values, and estimated treatment effects (Figure 3a,b). Pearson captures linear relationships, while Spearman reflects monotonic trends, offering complementary views of variable influence. Smoking pack-years showed the strongest and most consistent negative correlation across both matrices, reinforcing its role in reducing chemotherapy benefit. HPV positivity and younger age also exhibited modest positive correlations with SHAP values and effect estimates, aligning with known clinical patterns. Additional SHAP visualizations and discussion are provided in Appendix C.2.

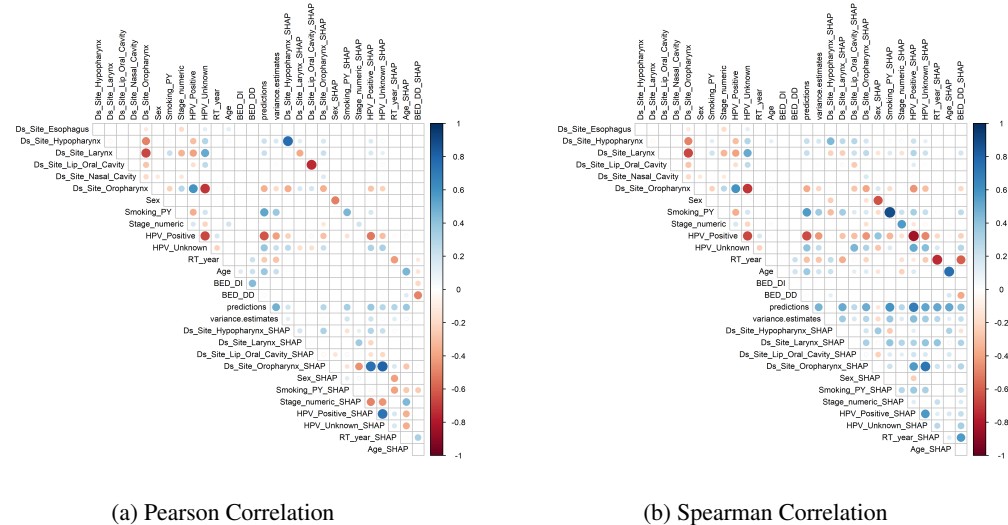

(a) Pearson Correlation            (b) Spearman Correlation

Figure 3: Correlation matrices between covariates, SHAP values, and treatment effects

**Robustness & Effect Heterogeneity:** CAST passed multiple validation checks, including dummy outcome tests, synthetic confounder experiments, and trimming sensitivity analyses. For the synthetic confounder tests, only the highest strengths of correlation with treatment assignment distorted the causal effect estimates dramatically, whereas smaller strengths had minimal impact. In the robustness checks with irrelevant features, as expected, the noise variables were largely ignored by the CSF

and did not substantially affect the causal effect estimates. Individualized treatment effect estimates exhibited a long right tail of high responders and a subset with near-zero or negative benefit. While some of this variation may reflect noise, the observed patterns indicate potential for personalized treatment modeling. Additional visualizations are provided in Appendix C.4.

## 6 Discussion

The patterns uncovered by CAST have important clinical implications. The observed peak in survival benefit around four to five years post-treatment suggests that chemotherapy is most effective for short to mid-term local control but may not sustain long-term survival. This decline could reflect tumor repopulation, distant progression, or delayed toxicity [51]. However, since fewer patients remained at risk (did not experience a death or censoring event) at longer follow-up times, reliability of the causal effect estimates at long times is reduced compared with shorter times, as shown by our dummy tests.

These findings support the value of adaptive monitoring and adjunct strategies to extend therapeutic benefit. The heterogeneity revealed by CAST emphasizes the need for treatment personalization. Correlation and SHAP-based analysis together identified HPV positivity and smoking as the most influential factors. Favorable outcomes in HPV-positive patients align with known radiosensitivity and impaired DNA repair, while smoking was linked to reduced benefit—consistent with mechanisms like tumor hypoxia and immunosuppression. Age also showed a modest effect, with younger patients generally benefiting more; an inflection point around 50–60 years may be clinically meaningful (Figure 3 and Figure 4 in Appendix C.2). In contrast, tumor site and TNM stage had limited influence on treatment effect heterogeneity, despite their prognostic relevance.

These findings align with efforts to tailor treatment by biologic subgroup. CAST offers a data-driven framework to support such stratifications and generate hypotheses for future trials. Rather than replacing existing tools, it complements them by modeling continuous-time dynamics and revealing patient-level variation. More broadly, this study shows how combining mechanistic modeling with causal machine learning can enhance the analysis of observational data. By embedding radiobiological insight into CAST using BED variants from different tumor repopulation models, we uncover treatment effects that align with known biology while also revealing discrepancies, such as stronger chemotherapy benefits than reported in prior meta-analyses. This offers a powerful way to complement clinical trials and generate new hypotheses.

**Limitations and Broader Impacts**

**Data limitations:** The dataset exhibits substantial right-censoring: while 88.9% of patients remain in follow-up at one year, only 22.2% do so by year six. This may bias long-term survival estimates and obscure treatment effects that manifest later in time. **External validity:** The data come from a single institution (University Health Network, Toronto) and are predominantly male (80%), limiting generalizability to broader populations, especially women. **Causal assumptions:** Like all causal inference methods, CAST relies on the assumption of no unmeasured confounding. Important factors such as diet, lifestyle, or genetic risk—potentially related to both treatment and outcome—are not included. **Methodological scope:** From a machine learning perspective, CAST supports only binary treatment variables. Extending it to model continuous dosing, multi-arm comparisons, or longitudinal interventions remains an important direction for future work.

## 7 Conclusion

In this paper, we present CAST, which is to our knowledge the first framework for modeling how treatment effects change over time using parametric and non-parametric techniques in the context of causal survival analysis with multiple features. CAST extends the utility of causal survival forests from estimating effects at discrete horizon times to continuous-time modeling. Applied to chemotherapy for HNSCC, CAST estimates individualized treatment trajectories and highlights when treatment effects peak and decline. Our results show that CAST is robust and interpretable, offering a general framework for modeling time-varying treatment effects across medical contexts. By isolating the causal influence of patient characteristics and capturing the dynamics of treatment response, CAST supports more personalized and adaptive care. This helps clinicians identify critical windows, tailor interventions to individual risk profiles, and refine strategies as new evidence emerges.

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

**Ethics Statement**

Existing at the intersection of machine learning (ML), healthcare, and causal inference, our work inevitably raises ethical considerations. By bringing ML methods to oncology research, we strive to advance personalized medicine and treatment strategies. However, our estimates are based on observational data and may be biased by unmeasured confounding. While the dataset includes a comprehensive description of variables including age, sex, smoking history, and HPV status, it omits race, ethnicity, and socioeconomic status data. These factors are key to understanding structural barriers to healthcare that could possibly affect outcomes. This risks amplifying existing biases in the data. ML models in oncology must be used cautiously and should not replace clinical judgment, but rather act as a supplement. Our findings require further clinical validation before integration into decision-making workflows.

## A Theoretical Justification of CAST

We provide formal justification for the consistency and identifiability of the time-varying treatment effect estimator $\hat{\tau}(t)$ used in the CAST framework.

### A.1 Problem Setting

Let $\mathcal{D} = \{(X_i, W_i, T_i, \delta_i)\}_{i=1}^n$ be a dataset of $n$ i.i.d. samples where: - $X_i \in \mathbb{R}^p$ is a vector of observed covariates, - $W_i \in \{0,1\}$ is a binary treatment indicator, - $T_i$ is the observed event or censoring time, - $\delta_i \in \{0,1\}$ is the event indicator (1 if the event occurred, 0 if censored).

Let $Y(w,t)$ denote the potential outcome (e.g., survival status at time $t$) under treatment $w \in \{0,1\}$.

We define the time-varying Conditional Average Treatment Effect (CATE) as:

$$\tau(x,t) := \mathbb{E}[Y(1,t) - Y(0,t) \mid X = x].$$

CAST estimates $\tau(x,t)$ using a doubly-robust causal survival forest followed by a spline or quadratic fit across time.

### A.2 Assumptions

We adopt standard causal inference and survival analysis assumptions:

(A1) **Unconfoundedness:** $(Y(0,t), Y(1,t)) \perp W \mid X$ for all $t$.

(A2) **Positivity:** $0 < P(W = 1 \mid X) < 1$ almost surely.

(A3) **Consistency:** $Y = Y(W,t)$ if $W$ is received.

(A4) **Non-informative Censoring:** $C \perp (Y(0,t), Y(1,t)) \mid X, W$ for censoring time $C$.

(A5) **Consistency of Forest Estimators:** The causal survival forests used yield consistent estimates of conditional survival functions $S_w(t \mid X)$.

### A.3 Theorem: Pointwise Consistency of $\hat{\tau}(t)$

[Pointwise Consistency] Under assumptions (A1)–(A5), for each fixed $t$:

$$\hat{\tau}(t) := \mathbb{E}_X[\hat{S}_1(t \mid X) - \hat{S}_0(t \mid X)] \xrightarrow{p} \tau(t) := \mathbb{E}_X[S_1(t \mid X) - S_0(t \mid X)]$$

as $n \to \infty$, where $\hat{S}_w(t \mid X)$ is the estimated conditional survival function under treatment $w$ from causal survival forests.

This follows from: 1. Consistency of $\hat{S}_w(t \mid X)$ (A5), 2. The continuous mapping theorem, since subtraction and expectation are continuous, 3. Trimming enforces overlap (A2), ensuring bounded inverse propensity weights.

### A.4 Identifiability of $\tau(t)$ from Observational Data

[Identifiability] Under assumptions (A1)–(A4), the marginal time-varying treatment effect

$$\tau(t) := \mathbb{E}_X[\mathbb{E}[Y \mid W = 1, X, T \geq t] - \mathbb{E}[Y \mid W = 0, X, T \geq t]]$$

is identified from observational data using inverse probability weighting or doubly-robust estimation.

Under unconfoundedness and non-informative censoring, we can consistently estimate the conditional means $\mathbb{E}[Y(w,t) \mid X]$ from observed data. The difference in conditional expectations across treatment groups yields an identifiable estimator of $\tau(t)$.

## A.5 Estimability of Peak Effect Time in CAST-Parametric

Let the parametric effect trajectory be:

$$\tau(t) = \beta_0 + \beta_1 t + \beta_2 t^2,$$

and suppose $\hat{\beta}_1, \hat{\beta}_2$ are estimated using weighted least squares.

[Consistency of Estimated Peak Time] If $\hat{\beta}_1 \xrightarrow{p} \beta_1$, $\hat{\beta}_2 \xrightarrow{p} \beta_2$ with $\beta_2 < 0$, then the estimated peak time

$$\hat{t}^* = -\frac{\hat{\beta}_1}{2\hat{\beta}_2}$$

is a consistent estimator of the true peak $t^* = -\frac{\beta_1}{2\beta_2}$.

This follows from Slutsky's theorem. Since both $\hat{\beta}_1$ and $\hat{\beta}_2$ converge in probability to non-zero limits, and the mapping $f(a, b) = -a/(2b)$ is continuous for $b \neq 0$, it follows that:

$$\hat{t}^* = -\frac{\hat{\beta}_1}{2\hat{\beta}_2} \xrightarrow{p} -\frac{\beta_1}{2\beta_2} = t^*.$$

## B Expanded Dataset Subsection

**Overview**

Our analysis uses the RADCURE dataset from The Cancer Imaging Archive (TCIA), the largest to our knowledge publicly accessible head and neck cancer imaging dataset. The data spans from 2005 to 2017 and includes computed tomography (CT) images for 3,346 patients, from which we selected a subset of 2,651 patients after filtering for only HNSCC cases. These images are linked to clinical, demographic, and treatment metadata. Following standardized clinical imaging protocols, the RAD-CURE project includes CT images, pictured alongside manually-reviewed contours differentiating between the planning tumor volume (PTV) and the organs at risk (OARs). All patients in this dataset received radiotherapy, and some received chemotherapy.

The clinical data accounts for patient demographics, including age, gender, and HPV status. It also details tumor staging using the 7th edition TNM system to describe the cancer, in addition to treatment information. While the dataset primarily focuses on oropharyngeal cancer, it also covers laryngeal, nasopharyngeal, and hypopharyngeal cancers.

**Data Preprocessing**

In the preprocessing stage, we filtered out incomplete patient profiles to ensure the dataset included relevant variables and appropriately represented potential confounders. We standardized all continuous variables to have zero mean and unit variance to ensure comparability and optimize model performance. The dataset comprehensively describes treatment details—dose/fraction, number of fractions, and total days of radiotherapy—which we used to calculate Biologically Effective Dose (BED) values. We implemented both dose-independent (DI) and dose-dependent (DD) BED models to capture the biological effects of radiation therapy, using established radiobiological parameters ($\alpha = 0.2\,\mathrm{Gy}^{-1}$, $\alpha/\beta = 10$ Gy, accelerated repopulation rates and onset times). This allowed us to quantify the effective radiation dose accounting for different fractionation schedules. We employed a stratified data partitioning strategy, creating training (75%) and testing (25%) sets while maintaining consistent event rates across partitions. Both subsets contained similar proportions of survival events, allowing for unbiased evaluation of treatment effects.

Table 1 summarizes the estimated average treatment effects across time for both restricted mean survival time (RMST) and survival probability (SP) metrics. These values were computed using causal survival forests on held-out test data. We observe that the estimated effects generally increase with longer follow-up, particularly under the RMST metric, reflecting the accumulating benefit of treatment over time. Standard errors are included to reflect model uncertainty at each horizon.

Table 1: Summary statistics of the simulated dataset

| Statistic | Control Group | Treated Group |
|---|---|---|
| Event Rate (%) | 79.8 | |
| Treatment Rate (%) | 44.9 | |
| Median Survival (months) | 17.0 | 24.0 |
| 12-month Survival (%) | 70.3 | 90.1 |
| 24-month Survival (%) | 20.2 | 45.5 |
| 36-month Survival (%) | 1.9 | 7.3 |
| 48-month Survival (%) | 0.0 | 0.1 |
| Age (mean) | 60.42 | 59.23 |
| TNM Stage (mean) | 1.73 | 3.46 |
| HPV Positivity Rate | 0.68 | 0.51 |
| Sex (Male = 1) | 0.48 | 0.49 |

**Computing Resources:** All experiments were conducted with a 13th Gen Intel Core i7-1355U CPU, 16GB RAM, and integrated Intel Iris Xe Graphics. No discrete GPU or cloud resources were used, though such resources would significantly reduce runtime for large-scale extensions of this work.

# C  Additional Results

In this section, we present additional results that extend and validate the findings reported in the main paper. These include visualizations of treatment effect heterogeneity across time, a summary of average treatment effects, and robustness checks to support the reliability of our causal estimates.

## C.1  Summary Table of Average Treatment Effects

Table 2 summarizes the estimated average treatment effects across time horizons using both RMST and survival probability metrics. These values were computed using causal survival forests on the held-out test set. The treatment effects tend to increase over time under both metrics, with RMST showing a steeper upward trend reflecting cumulative benefit. Standard errors are included for each estimate. The early rise in both SP and RMST suggests initial treatment efficacy, while the plateauing in later months reflects diminishing returns, possibly due to recurrence or late toxicity. The RMST gains—peaking at over 16 months—highlight how cumulative survival benefit continues to accrue even as survival probability differences taper off. These patterns support the biological intuition that treatment effects rise quickly post-intervention and then gradually attenuate.

Table 2: Estimated average treatment effects (ATE) across time using RMST and survival probability (SP). SE represent standard errors

| Months | ATE (SP) | SE (SP) | ATE (RMST) | SE (RMST) |
|--------|----------|---------|------------|-----------|
| 12 | 0.099 | 0.049 | 0.44 | 0.26 |
| 24 | 0.141 | 0.053 | 1.88 | 0.80 |
| 36 | 0.152 | 0.058 | 3.58 | 1.46 |
| 48 | 0.178 | 0.072 | 5.80 | 2.31 |
| 60 | 0.168 | 0.071 | 7.39 | 2.73 |
| 72 | 0.148 | 0.075 | 8.38 | 3.52 |
| 84 | 0.156 | 0.077 | 11.08 | 4.76 |
| 96 | 0.143 | 0.071 | 13.89 | 5.90 |
| 108 | 0.129 | 0.068 | 14.76 | 6.16 |
| 120 | 0.100 | 0.063 | 16.11 | 6.92 |

These summary statistics also inform the CAST modeling strategies described in Section 3.3. The steady increase followed by tapering motivates the use of both quadratic and spline-based approaches to flexibly capture the full temporal arc of treatment efficacy.

## C.2    SHAP-Based Interpretability Analysis

While SHAP provides valuable insights into feature influence, the estimates generated here using the fastshap R package are approximate and may be noisy, particularly in the context of survival analysis. We calculated approximate SHAP values because an exact SHAP explainer does not yet exist for the causal survival forest model. Figures 4(a–c) show SHAP plots for the three most influential variables—age, HPV status, and smoking pack-years—highlighting clear heterogeneity in treatment benefit across subgroups. Additional SHAP plots for other covariates—such as tumor site, treatment timing, dose metrics, and TNM stage—are also provided below. These variables had smaller contributions to the model, but are shown for completeness and transparency.

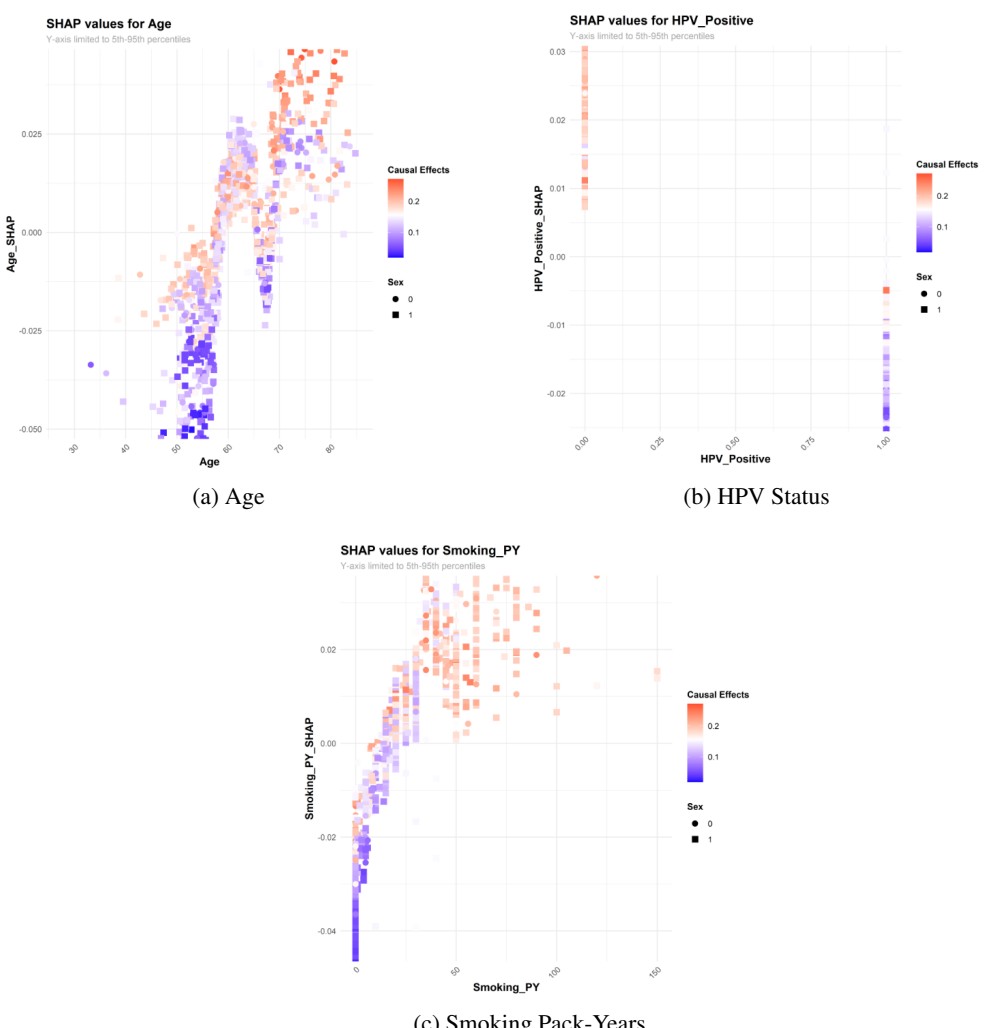

(a) Age

(b) HPV Status

(c) Smoking Pack-Years

Figure 4: SHAP analysis of covariates driving treatment effect heterogeneity. (a) Older age is linked to greater chemotherapy benefit. (b) HPV-negative patients consistently show higher contributions. (c) Smoking history is positively associated with the chemotherapy benefit treatment.

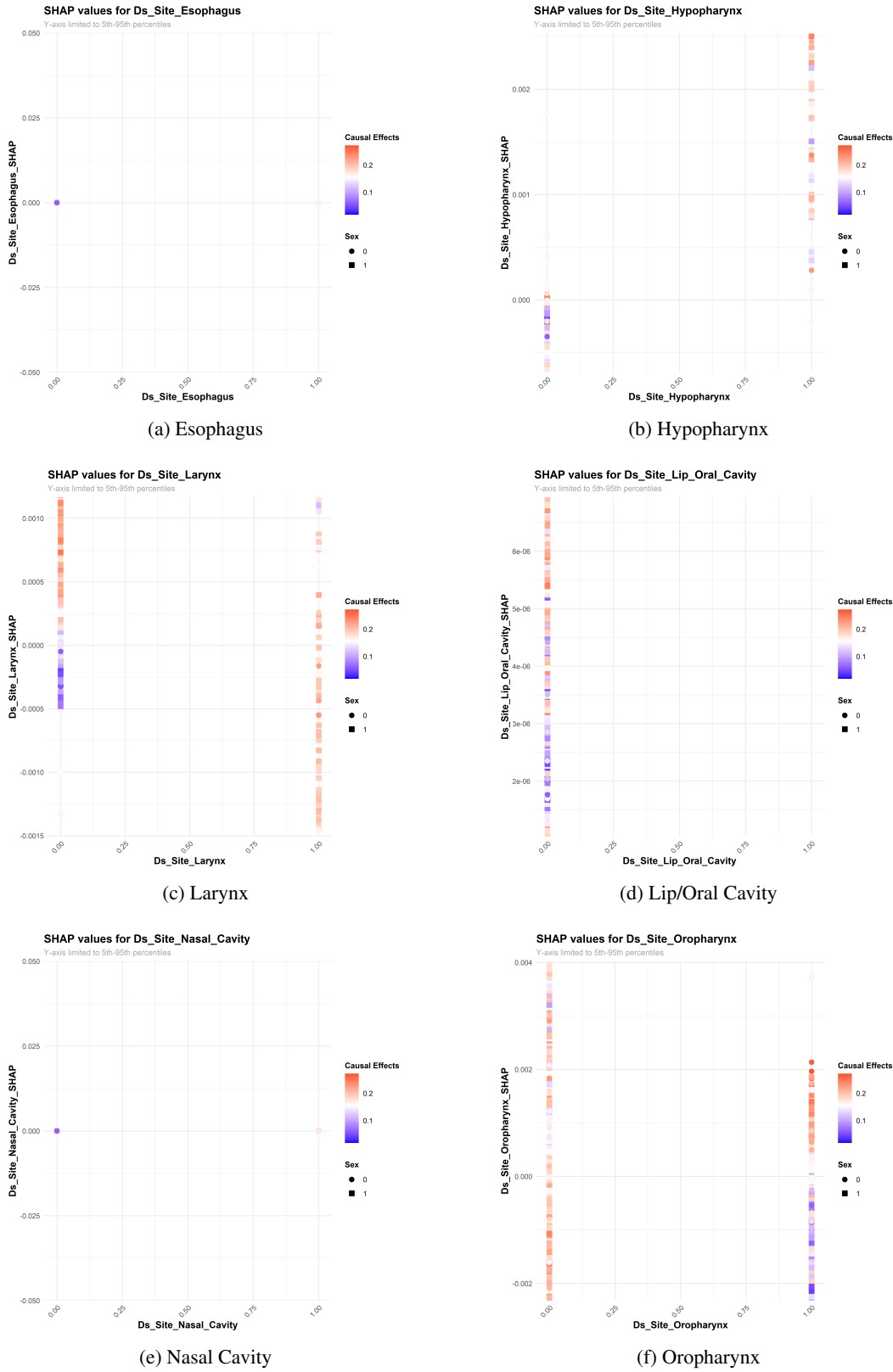

Figure 5: SHAP values for primary tumor site. These anatomical subgroups exhibited low or diffuse contributions to treatment effect heterogeneity, though subtle site-specific trends may still hold clinical value.

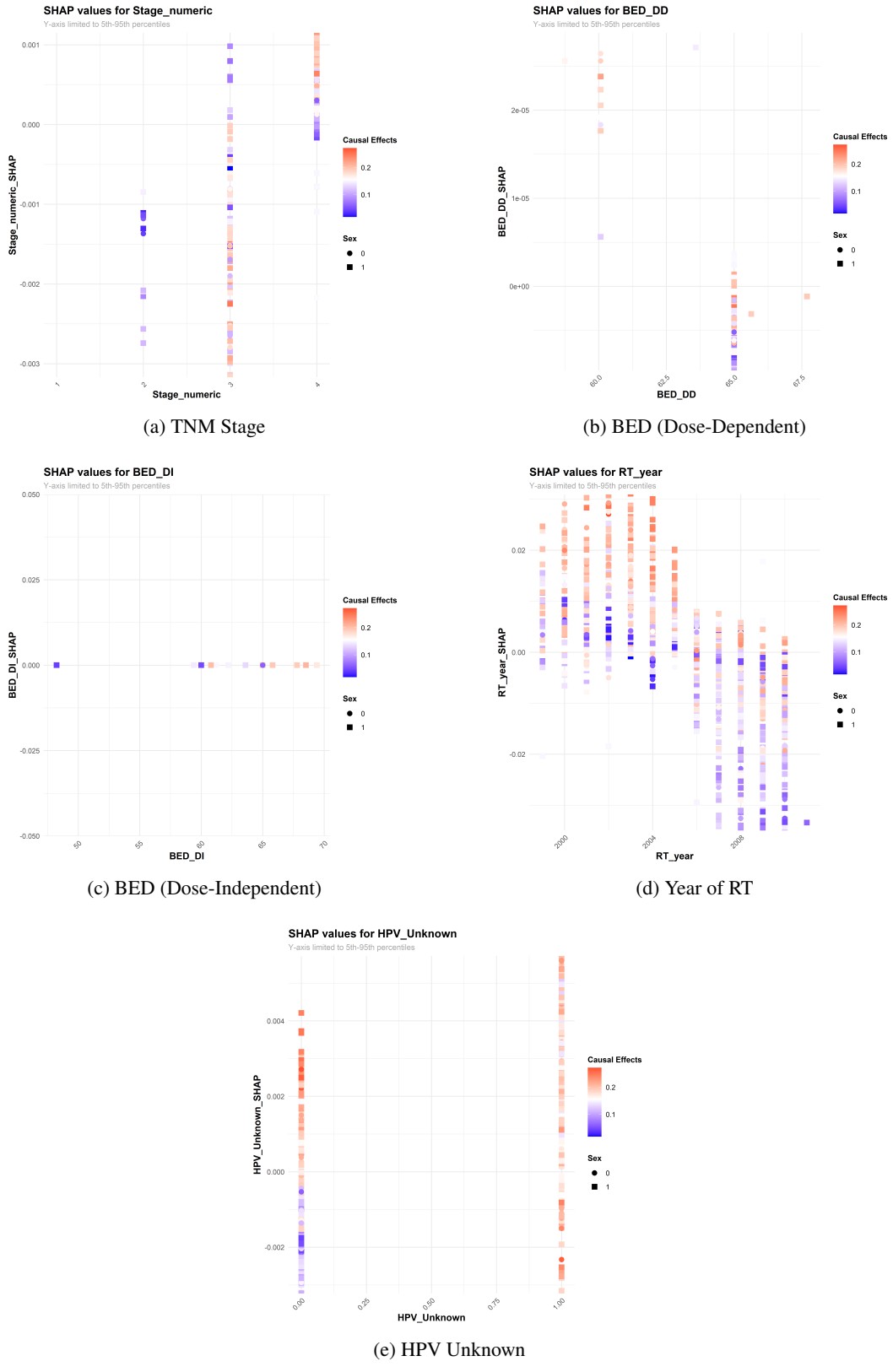

Figure 6: SHAP values for additional covariates, including TNM stage, treatment year, and dose-related metrics. These features showed limited or context-specific contributions to treatment effect heterogeneity.

## C.3 Distributions of Individualized Treatment Effects

We visualize the estimated treatment effect distributions for both RMST and survival probability (SP) at intervals ranging from 12 to 120 months. Figures 4 and 5 show individual-level causal effects derived from the causal survival forest at each time horizon.

### RMST Treatment Effect Distributions

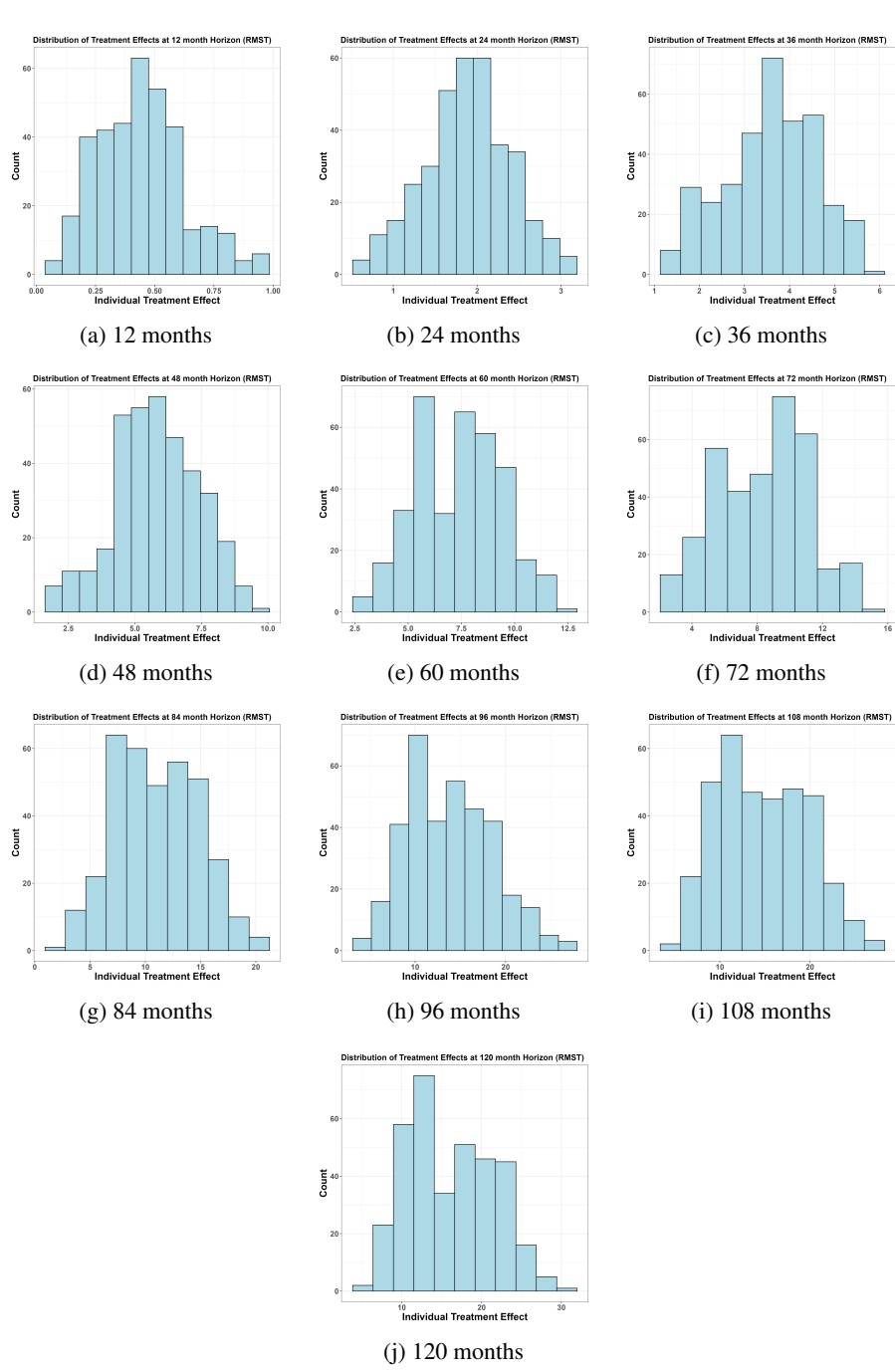

Figure 7: Distributions of estimated RMST-based treatment effects over time. Each panel shows the individual-level causal effect at a specific horizon as learned by the causal survival forest.

## Survival Probability Treatment Effect Distributions

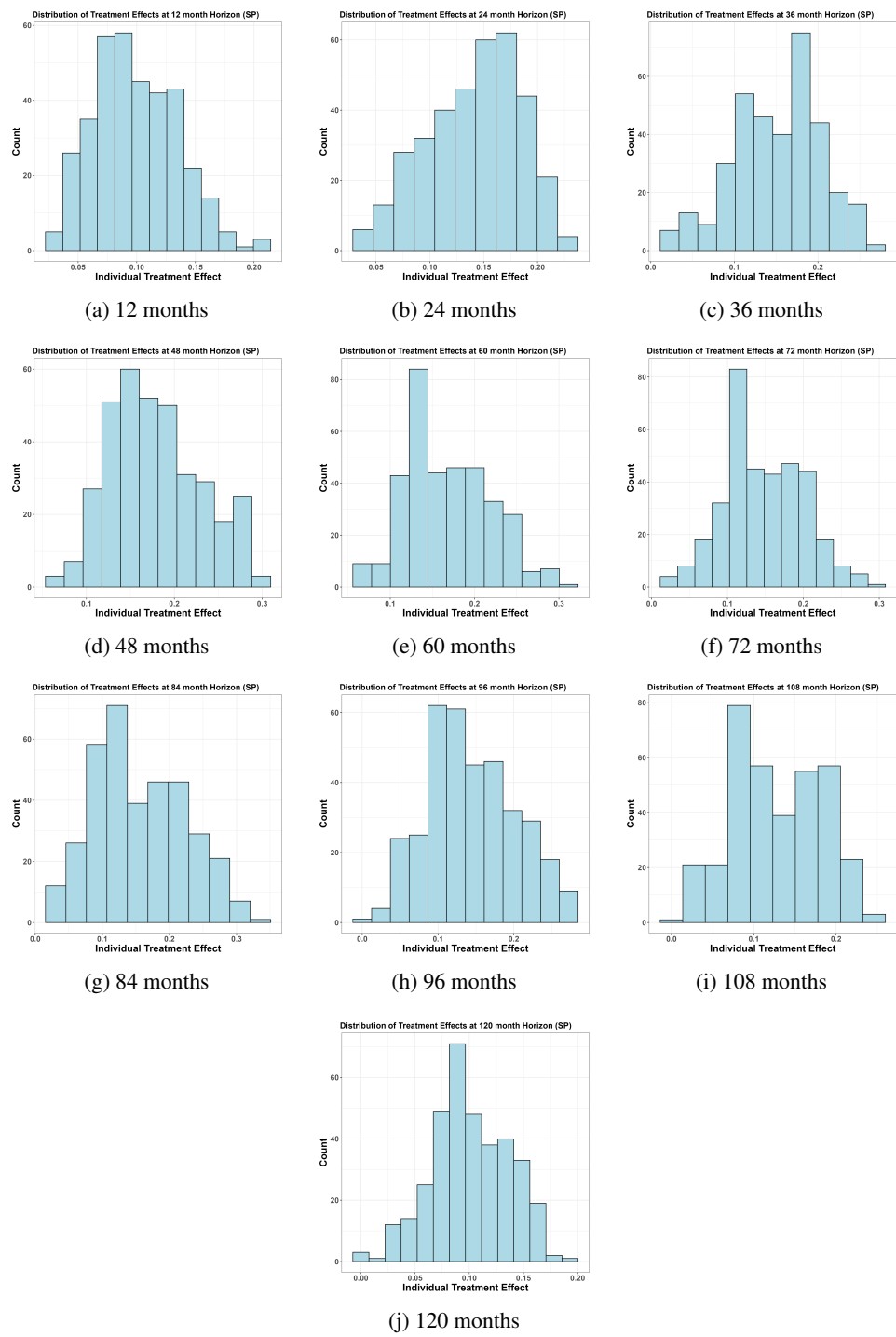

Figure 8: Distributions of estimated survival-probability-based treatment effects over time. Each panel shows the individual-level causal effect at a specific horizon as estimated by the causal survival forest.

## C.4 Dummy Outcome Refutation Tests

To assess whether CAST detects spurious treatment effects in the absence of a true signal, we performed dummy outcome tests. For each time horizon, we randomly shuffled treatment assignments and outcome times across 20 repetitions to simulate a null setting. If the model was overfitting or improperly attributing causal structure, it would produce non-zero treatment effect estimates even under randomization. As shown in the boxplots below, the estimated treatment effects for both RMST and survival probability are centered around zero, especially at relatively short times ($\leq 60$ months), when the number of patients still at risk was large. This confirms that CAST does not learn artifacts from the data and is robust to randomization of causal structure.

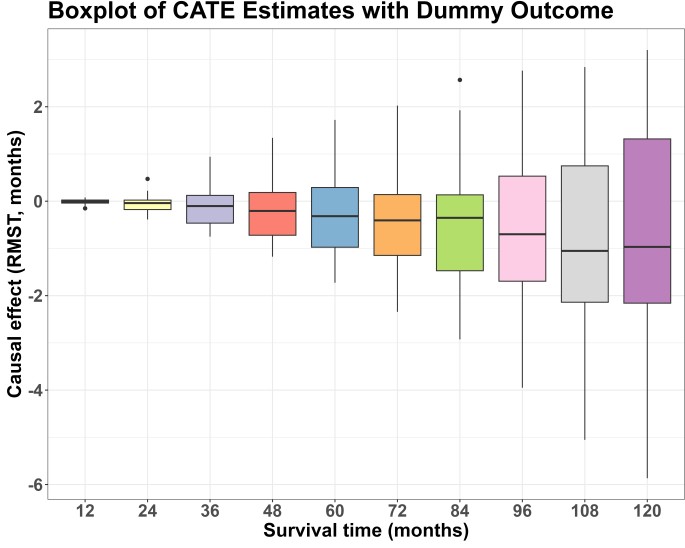

Figure 9: Dummy outcome test for RMST-based ATE estimates. Across 20 shuffles per horizon, treatment effects are centered near zero, consistent with the null.

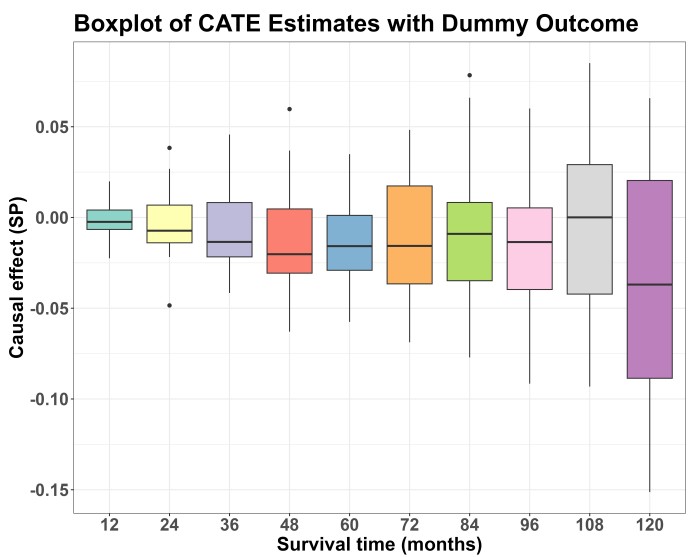

Figure 10: Dummy outcome test for survival probability-based ATE estimates. The model correctly reports no significant treatment effects under randomized labels.

To assess the robustness of CAST estimates to unobserved confounding, we performed a sensitivity analysis by injecting synthetic covariates with varying correlation to treatment assignment ($r = 0.1$, 0.3, 0.5). We then measured the resulting shifts in ATE estimates across time horizons for both RMST and survival probability outcomes.

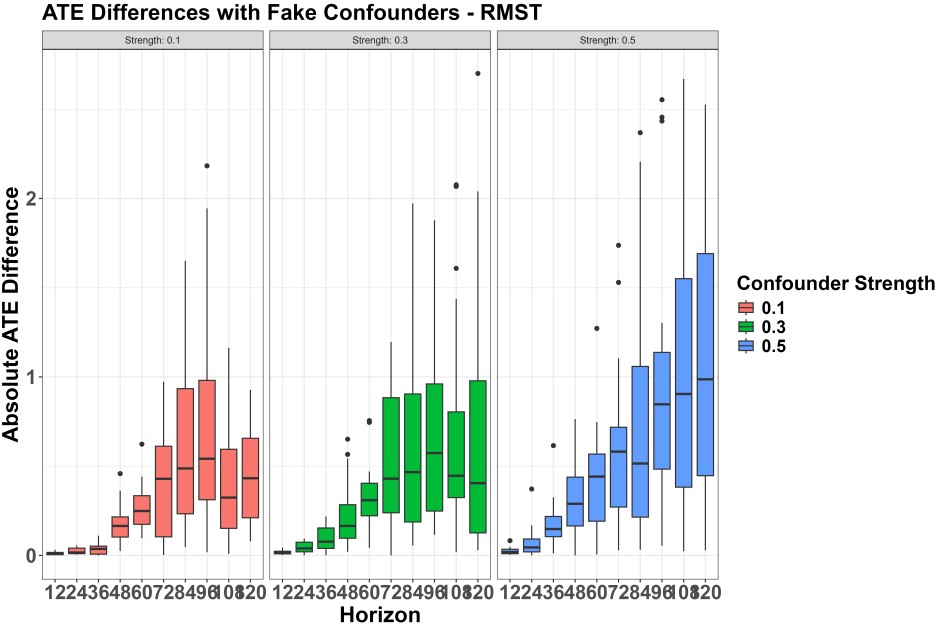

Figure 11: Absolute ATE differences in RMST under varying confounder strengths ($r = 0.1$, 0.3, 0.5). Estimates are stable under weak strengths but diverge at longer horizons and higher strengths.

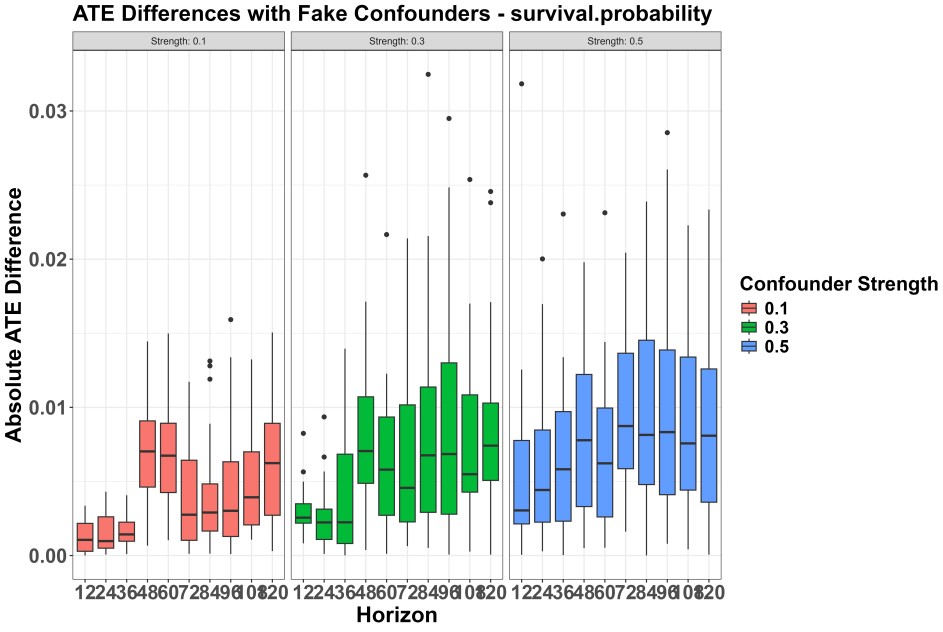

Figure 12: Absolute ATE differences in SP under varying confounder strengths ($r = 0.1$, 0.3, 0.5). CAST estimates remain stable under weak strengths, with modest shifts at stronger levels and longer horizons.

