# OpenReview forum: "CAST: Time-Varying Treatment Effects with Application to Chemotherapy and Radiotherapy on Head and Neck Squamous Cell Carcinoma"
_NeurIPS.cc/2025/Conference — Submitted to NeurIPS 2025_

### Official Review · Reviewer_LLGh · 2025-06-03

**Clarity:** 4
**Significance:** 2
**Originality:** 1
**Rating:** 2
**Confidence:** 4

**Summary:**

The paper proposes CAST (Causal analysis for survival trajectories) for estimating treatment effects as functions of time after treatment. Specifically, CAST is a continuous time method that overcomes modeling limitations of discrete time methods. Through a real-word case study on the RADCURE dataset, the authors provide insights on the dynamics of radio and  chemotherapy for patients with head and neck squamous cell carcinoma using their proposed method.

**Questions:**

- Can the authors explain the technical novelty in their proposed approach?
- Can the authors report the performance of CAST against baselines on additional datasets?

**Ethical Concerns:**

["NO or VERY MINOR ethics concerns only"]

**Limitations:**

The authors discuss limitations regarding the dataset, but no limitations of the validity of their causal assumptions nor of CAST itself are discussed.

**Paper Formatting Concerns:**

no formatting concerns

**Quality:**

2

**Strengths And Weaknesses:**

Strengths:
- The topic is relevant.
- The paper is well written. It is easy to follow and nice to read.
- The evaluation study on the RADCURE dataset provides valuable insights.

Weaknesses:
- No technical contribution: The method is very straight forward. The proposed algorithms are not novel but simply employ weighted least squares  / smoothing splines. There is no technical contribution.
- Limited evaluation: While the evaluation on the RADCURE dataset is extensive, there should be some sort of benchmarking against baseline methods, and further performance studies on additional datasets (e.g., synthetic data). This way, it is hard to tell whether the performance on RADCURE is simply due to a well-chosen dataset.
- Related work: There are many papers on heterogeneous and average treatment effect estimation in the continuous time setting [e.g., 1,2,3,4,5]. While this is a slightly different setting, it may be worth mentioning.

I believe the paper has potential, but rather for a more applied outlet (e.g., ML4H). Due to the limited technical contribution, I vote for reject.

Minor:
- Line 74: Fig. 1 should have a reference to the figure.
- Line 301: Fig. 2 should have a reference to the figure.
- Lines 77-84: Past tense instead of present tense.
- I would move the paragraph in Lines 143-147 to the related work.
- Lines 162-165 are implementation details, I would put them somewhere else.
___
[1] Konstantin Hess, and Stefan Feuerriegel. Stabilized neural prediction of potential outcomes in continuous time. In ICLR, 2025.

[2] Kjetil Røysland. A martingale approach to continuous-time marginal structural models. Bernoulli, 17(3):895 – 915, 2011.

[3] Helene C. Rytgaard, Thomas A. Gerds, and Mark J. van der Laan. Continuous-time targeted minimum loss-based estimation of intervention-specific mean outcomes. The Annals of Statistics, 2022.

[4] Helene C. Rytgaard, Frank Eriksson, and Mark J van der Laan. Estimation of time-specific intervention effects on continuously distributed time-to-event outcomes by targeted maximum likelihood estimation. Biometrics, 79(4):3038–3049, 2023.

[5] Nabeel Seedat, Fergus Imrie, Alexis Bellot, Zhaozhi Qian, and Mihaela van der Schaar. Continuous-time modeling of counterfactual outcomes using neural controlled differential equations. In ICML, 2022.

---

### Official Review · Reviewer_ofQi · 2025-06-26

**Clarity:** 3
**Significance:** 2
**Originality:** 1
**Rating:** 2
**Confidence:** 4

**Summary:**

The authors develop a causal machine learning framework, called causal analysis for survival forests estimates (CAST), that models treatment effects over time with the use of algorithms involving parametric function and non-parametric smoothing. From the parametric functions, one can estimate a series of parameters that can guide clinicians on the peak treatment effect, when that might occur, and when that might wane. From the non-parametric function, one can determine inflection of points of the time-varying treatment effect.

**Questions:**

Some aspects of this methodology, which is very straightforward, are unique. However, the authors do seem to ignore some of the more recent research in time-varying treatment effects within G-formula and double-robust methodology, more specifically targeted maximum likelihood estimation, that have incorporated ML algorithms for this very purpose. These methodologies have also confronted several mechanisms of bias that are not addressed with CAST (e.g., informative censoring, collider and stratification bias, etc.).

The parametric and non-parametric approaches are extremely helpful in this application and I completely agree they complement one another (ln 212). One major issue is that the parametric component only involves a quadratic function and it should be extended to other functions. Such an extension would require a reassessment of the t_peak and half-life lambda estimation in Algorithm 1.

There could be instances where there is no time-varying treatment effect, thereby posing several downstream consequences for this framework. In the current parametric set-up, if beta_2 = 0, then half-life cannot be estimated (the “degenerate case” in algorithm 1). beta_1 = 0 and beta_2 = 0 could also be a possibility (although mathematically redundant for algorithm 1), but brings up the possibility that there are no t_peak and tau(t_peak) to estimate simply because the function is not quadratic. In the non-parametric set-up, the inflection points are determined from numerically-derived first and second derivatives of the smoothed spline function. There could be instances where there are multiple inflection points (for more complex functions) or no infection points (in the absence of time-varying treatment effects), making inflection points less interesting. Algorithm 2 does not seem to handle this (renders to 0 or NA?). In the application the authors also state that the effects are stable from the dummy analysis, yet propose limited alternatives (ln 304-305).

There are some questions with interpretability of the output. t_peak is probably a function of RMST (or some other survival component), so it might not be of interest. tau(t_peak) is probably more interesting in that it gives the largest possible treatment benefit over time, but could be compared with a minimal treatment estimate, i.e., tau(t_min), answering whether the maximum effect is truly that maximal. How quickly the effect diminishes, i.e., half-life, is not particularly useful for clinicians, as this time will never change when treatment will be restarted or screening will be intensified, for example. Perhaps if this information were linked to recurrent cancer, the usefulness of this endpoint could be justified.

In terms of treatment, it is easier to determine covariates involved in prediction of commencing a cancer treatment regimen in first-line therapy than subsequent therapies. If first line therapy, the application with PS is likely adequate. If there are multiple rounds of therapy, then treatment W_i is going to be conditional on the preceding treatment regimen and the success of those treatments. Can these be incorporated in the current framework?


There is no discussion of informative censoring, which has severe consequences on estimating treatment effects. There will be covariates that influence T_i or censoring in general, such as C_i in the G-formula based methods described in ref 39. The authors claim to have properly accounted for right-censoring, but it is unclear how this was accomplished from refs 49 and 50. Dummy tests also show the reliability of the causal estimates are reduced (lines 337-338).

Lastly, those who die in the beginning of follow-up are, counterfactually, likely to die regardless of which treatment was given. Including these individuals will bias the causal effect and the consistency assumptions (mentioned in lns 159-160) might be jeopardized.


Minor comments:
- ln 43-44, 335-336. The authors discuss how their methodology might relate to complex temporal dynamics of tumor control/toxicities. Given that all parameters are essentially based on time after treatment discontinuation, the authors could be clearer about this link.
- ln 51-55. Rising incidence of HPV-related cases of head and neck cancers (especially among younger populations) might be exaggerated here, given that the generations who received HPV vaccination as part of national vaccination programs have entered mid adulthood.
- ln 80-81. The trimming (removing patients from the analysis?) of the propensity scores to [0.1, 0.9] does not guarantee that the positivity assumption has been met.
- ln 233. Hypothesis testing is stated here, but to my knowledge has not been carried out in this research. The authors could expand on this, as I imagine hypothesis testing would be done on t_peak, tau(t_peak) and half-life lambda.
- ln 247-248. What exactly is the treatment modeled? A binary indicator is given but dose/fraction, number of fractions and total radiation treatment time suggests varying treatment regime (SUTVA assumption is unlikely if there is too much variation in treatment regimes).
- ln 273, It is unclear is time-varying covariates are also being handled with the SHAP_i estimates. Indeed, the covariates in this exercise are clearly time-stable, but other applications might require SHAP_i for more dynamic settings.
- Figure 1. Missing y-axis.

**Ethical Concerns:**

["NO or VERY MINOR ethics concerns only"]

**Limitations:**

Some aspects of this methodology, which is very straightforward, are unique. However, the authors do seem to ignore some of the more recent research in time-varying treatment effects within G-formula and double-robust methodology, more specifically targeted maximum likelihood estimation, that have incorporated ML algorithms for this very purpose. These methodologies have also confronted several mechanisms of bias that are not addressed with CAST (e.g., informative censoring, collider and stratification bias, etc.).

The parametric and non-parametric approaches are extremely helpful in this application and I completely agree they complement one another (ln 212). One major issue is that the parametric component only involves a quadratic function and it should be extended to other functions. Such an extension would require a reassessment of the t_peak and half-life lambda estimation in Algorithm 1.

There could be instances where there is no time-varying treatment effect, thereby posing several downstream consequences for this framework. In the current parametric set-up, if beta_2 = 0, then half-life cannot be estimated (the “degenerate case” in algorithm 1). beta_1 = 0 and beta_2 = 0 could also be a possibility (although mathematically redundant for algorithm 1), but brings up the possibility that there are no t_peak and tau(t_peak) to estimate simply because the function is not quadratic. In the non-parametric set-up, the inflection points are determined from numerically-derived first and second derivatives of the smoothed spline function. There could be instances where there are multiple inflection points (for more complex functions) or no infection points (in the absence of time-varying treatment effects), making inflection points less interesting. Algorithm 2 does not seem to handle this (renders to 0 or NA?). In the application the authors also state that the effects are stable from the dummy analysis, yet propose limited alternatives (ln 304-305).

There are some questions with interpretability of the output. t_peak is probably a function of RMST (or some other survival component), so it might not be of interest. tau(t_peak) is probably more interesting in that it gives the largest possible treatment benefit over time, but could be compared with a minimal treatment estimate, i.e., tau(t_min), answering whether the maximum effect is truly that maximal. How quickly the effect diminishes, i.e., half-life, is not particularly useful for clinicians, as this time will never change when treatment will be restarted or screening will be intensified, for example. Perhaps if this information were linked to recurrent cancer, the usefulness of this endpoint could be justified.

In terms of treatment, it is easier to determine covariates involved in prediction of commencing a cancer treatment regimen in first-line therapy than subsequent therapies. If first line therapy, the application with PS is likely adequate. If there are multiple rounds of therapy, then treatment W_i is going to be conditional on the preceding treatment regimen and the success of those treatments. Can these be incorporated in the current framework?


There is no discussion of informative censoring, which has severe consequences on estimating treatment effects. There will be covariates that influence T_i or censoring in general, such as C_i in the G-formula based methods described in ref 39. The authors claim to have properly accounted for right-censoring, but it is unclear how this was accomplished from refs 49 and 50. Dummy tests also show the reliability of the causal estimates are reduced (lines 337-338).

Lastly, those who die in the beginning of follow-up are, counterfactually, likely to die regardless of which treatment was given. Including these individuals will bias the causal effect and the consistency assumptions (mentioned in lns 159-160) might be jeopardized.


Minor comments:
- ln 43-44, 335-336. The authors discuss how their methodology might relate to complex temporal dynamics of tumor control/toxicities. Given that all parameters are essentially based on time after treatment discontinuation, the authors could be clearer about this link.
- ln 51-55. Rising incidence of HPV-related cases of head and neck cancers (especially among younger populations) might be exaggerated here, given that the generations who received HPV vaccination as part of national vaccination programs have entered mid adulthood.
- ln 80-81. The trimming (removing patients from the analysis?) of the propensity scores to [0.1, 0.9] does not guarantee that the positivity assumption has been met.
- ln 233. Hypothesis testing is stated here, but to my knowledge has not been carried out in this research. The authors could expand on this, as I imagine hypothesis testing would be done on t_peak, tau(t_peak) and half-life lambda.
- ln 247-248. What exactly is the treatment modeled? A binary indicator is given but dose/fraction, number of fractions and total radiation treatment time suggests varying treatment regime (SUTVA assumption is unlikely if there is too much variation in treatment regimes).
- ln 273, It is unclear is time-varying covariates are also being handled with the SHAP_i estimates. Indeed, the covariates in this exercise are clearly time-stable, but other applications might require SHAP_i for more dynamic settings.
- Figure 1. Missing y-axis.

**Paper Formatting Concerns:**

See above

**Quality:**

2

**Strengths And Weaknesses:**

Some aspects of this methodology, which is very straightforward, are unique. However, the authors do seem to ignore some of the more recent research in time-varying treatment effects within G-formula and double-robust methodology, more specifically targeted maximum likelihood estimation, that have incorporated ML algorithms for this very purpose. These methodologies have also confronted several mechanisms of bias that are not addressed with CAST (e.g., informative censoring, collider and stratification bias, etc.).

The parametric and non-parametric approaches are extremely helpful in this application and I completely agree they complement one another (ln 212). One major issue is that the parametric component only involves a quadratic function and it should be extended to other functions. Such an extension would require a reassessment of the t_peak and half-life lambda estimation in Algorithm 1.

There could be instances where there is no time-varying treatment effect, thereby posing several downstream consequences for this framework. In the current parametric set-up, if beta_2 = 0, then half-life cannot be estimated (the “degenerate case” in algorithm 1). beta_1 = 0 and beta_2 = 0 could also be a possibility (although mathematically redundant for algorithm 1), but brings up the possibility that there are no t_peak and tau(t_peak) to estimate simply because the function is not quadratic. In the non-parametric set-up, the inflection points are determined from numerically-derived first and second derivatives of the smoothed spline function. There could be instances where there are multiple inflection points (for more complex functions) or no infection points (in the absence of time-varying treatment effects), making inflection points less interesting. Algorithm 2 does not seem to handle this (renders to 0 or NA?). In the application the authors also state that the effects are stable from the dummy analysis, yet propose limited alternatives (ln 304-305).

There are some questions with interpretability of the output. t_peak is probably a function of RMST (or some other survival component), so it might not be of interest. tau(t_peak) is probably more interesting in that it gives the largest possible treatment benefit over time, but could be compared with a minimal treatment estimate, i.e., tau(t_min), answering whether the maximum effect is truly that maximal. How quickly the effect diminishes, i.e., half-life, is not particularly useful for clinicians, as this time will never change when treatment will be restarted or screening will be intensified, for example. Perhaps if this information were linked to recurrent cancer, the usefulness of this endpoint could be justified.

In terms of treatment, it is easier to determine covariates involved in prediction of commencing a cancer treatment regimen in first-line therapy than subsequent therapies. If first line therapy, the application with PS is likely adequate. If there are multiple rounds of therapy, then treatment W_i is going to be conditional on the preceding treatment regimen and the success of those treatments. Can these be incorporated in the current framework?

There is no discussion of informative censoring, which has severe consequences on estimating treatment effects. There will be covariates that influence T_i or censoring in general, such as C_i in the G-formula based methods described in ref 39. The authors claim to have properly accounted for right-censoring, but it is unclear how this was accomplished from refs 49 and 50. Dummy tests also show the reliability of the causal estimates are reduced (lines 337-338).

Lastly, those who die in the beginning of follow-up are, counterfactually, likely to die regardless of which treatment was given. Including these individuals will bias the causal effect and the consistency assumptions (mentioned in lns 159-160) might be jeopardized.


Minor comments:
- ln 43-44, 335-336. The authors discuss how their methodology might relate to complex temporal dynamics of tumor control/toxicities. Given that all parameters are essentially based on time after treatment discontinuation, the authors could be clearer about this link.
- ln 51-55. Rising incidence of HPV-related cases of head and neck cancers (especially among younger populations) might be exaggerated here, given that the generations who received HPV vaccination as part of national vaccination programs have entered mid adulthood.
- ln 80-81. The trimming (removing patients from the analysis?) of the propensity scores to [0.1, 0.9] does not guarantee that the positivity assumption has been met.
- ln 233. Hypothesis testing is stated here, but to my knowledge has not been carried out in this research. The authors could expand on this, as I imagine hypothesis testing would be done on t_peak, tau(t_peak) and half-life lambda.
- ln 247-248. What exactly is the treatment modeled? A binary indicator is given but dose/fraction, number of fractions and total radiation treatment time suggests varying treatment regime (SUTVA assumption is unlikely if there is too much variation in treatment regimes).
- ln 273, It is unclear is time-varying covariates are also being handled with the SHAP_i estimates. Indeed, the covariates in this exercise are clearly time-stable, but other applications might require SHAP_i for more dynamic settings.
- Figure 1. Missing y-axis.

---

### Official Review · Reviewer_EHVd · 2025-06-29

**Clarity:** 2
**Significance:** 2
**Originality:** 2
**Rating:** 3
**Confidence:** 2

**Summary:**

This paper proposes Causal Analysis for Survival Trajectories (CAST), a framework that models treatment effects as continuous functions of time after intervention. By integrating both parametric and non-parametric approaches, CAST addresses the limitations of discrete time-point analyses and enables the estimation of continuous effect trajectories. The authors apply CAST to the RADCURE dataset, demonstrating its ability to capture the evolving effects of chemotherapy and radiotherapy over time at both the population and individual levels.

**Questions:**

Could the author explain the difference between CAST and following papers:

1. Cao, Defu, James Enouen, and Yan Liu. "Estimating Treatment Effects in Continuous Time with Hidden Confounders."
2. Zhang, Mingyuan, Marshall M. Joffe, and Dylan S. Small. "Causal inference for continuous-time processes when covariates are observed only at discrete times." Annals of statistics 39.1 (2011): 10-1214.
3. Lok, Judith J. "Statistical modeling of causal effects in continuous time." (2008): 1464-1507.

**Ethical Concerns:**

["NO or VERY MINOR ethics concerns only"]

**Final Justification:**

With no provided rebuttal, I will keep my initial assessment.

**Limitations:**

As I mentioned previously,

1. The method is evaluated on only a single dataset. Incorporating additional datasets would strengthen the generalizability and robustness of the results.

2. A simulation study would be valuable to assess CAST’s performance under controlled settings and varying assumptions.

3. The paper would be more compelling with a direct comparison between CAST and existing causal effect estimation methods to better contextualize its advantages.

**Quality:**

2

**Strengths And Weaknesses:**

Strength: This paper aims to address an important problem in continuous time causal effect estimation and survival analysis. The proposed CAST framework is evaluated on a real-world dataset, demonstrating its practical utility and relevance to clinical applications.

Weakness

1. The method is evaluated on only a single dataset. Incorporating additional datasets would strengthen the generalizability and robustness of the results.

2. A simulation study would be valuable to assess CAST’s performance under controlled settings and varying assumptions.

3. The paper would be more compelling with a direct comparison between CAST and existing causal effect estimation methods to better contextualize its advantages.

---

### Official Review · Reviewer_f4WL · 2025-07-02

**Clarity:** 2
**Significance:** 1
**Originality:** 1
**Rating:** 2
**Confidence:** 4

**Summary:**

The paper extends discrete-time treatment effect predictions from causal survival forests (CSF) to continuous time by interpolating predictions across independently trained CSF models over pre-specified time horizons, using both a parametric quadratic function and a non-parametric quadratic spline function. Experimental results on head and neck squamous cell carcinoma (HNSC) contrast the estimated continuous-time differences in probabilities between parametric and non-parametric functions, identify covariates that result in heterogeneous outcomes, and evaluate the robustness of effect estimations.

**Questions:**

- Could you benchmark the work on additional datasets and include results from other related works, such as [1] and [2], and provide hazard ratio estimands?

**Ethical Concerns:**

["NO or VERY MINOR ethics concerns only"]

**Limitations:**

- The paper should discuss how trimming the propensity scores could introduce bias into their modeling approach.
- The paper should address the robustness results, which demonstrate that the treatment effect predictions worsen over longer time horizons, and disentangle the effects of modeling the CSF independently across all time horizons.
- The paper should also discuss the impact of the pre-specified time horizon on the treatment effect estimation.

**Quality:**

1

**Strengths And Weaknesses:**

**Strengths**

- The paper focuses on the continuous-time survival heterogeneous treatment effect estimation problem, which is an important but under-explored area.
- The robustness of the effect estimation experiments are interesting.

**Weaknesses**
- The paper seems to be a straightforward extension of the CSF model to continuous-time prediction using quadratic parametric or non-parametric function smoothing methods.
- The use of independent CSF models for different time horizons seems flawed. This is also evident from the robustness experiments, which demonstrate that the predictions are not robust over longer time horizons under the validation checks.
- The paper is missing several important benchmarks, such as [1], which estimates non-parametric continuous-time treatment effects, and [2], a propensity-weighted CoxPH model that enables estimation of covariate-specific hazard ratios.
- The paper should also include the hazard ratio estimand, which is considered more informative.
- Given that the technical contributions of this work are limited, the paper should include additional RCT and non-RCT datasets to evaluate the effectiveness of their continuous-time modeling approach.

**References**

- [1] Chapfuwa et al., "Enabling counterfactual survival analysis with balanced representations." Proceedings of the Conference on Health, Inference, and Learning, 2021.
- [2] Schemper et al., "The estimation of average hazard ratios by weighted Cox regression," Statistics in Medicine, 2009.

---

### Decision · Program_Chairs · 2025-09-17

**Decision:**

Reject

**Comment:**

Authors did not provide any rebuttal and all reviewers gave initial low ratings